# Unnatural language processing:
# How do language models handle machine-generated prompts?

**Corentin Kervadec** and **Francesca Franzon**
Universitat Pompeu Fabra (UPF) / Barcelona
{name.lastname}@upf.edu

**Marco Baroni**
UPF and ICREA / Barcelona
marco.baroni@upf.edu

## Abstract

Language model prompt optimization research has shown that semantically and grammatically well-formed manually crafted prompts are routinely outperformed by automatically generated token sequences with no apparent meaning or syntactic structure, including sequences of vectors from a model's embedding space. We use machine-generated prompts to probe how models respond to input that is not composed of natural language expressions. We study the behavior of models of different sizes in multiple semantic tasks in response to both continuous and discrete machine-generated prompts, and compare it to the behavior in response to human-generated natural-language prompts. Even when producing a similar output, machine-generated and human prompts trigger different response patterns through the network processing pathways, including different perplexities, different attention and output entropy distributions, and different unit activation profiles. We provide preliminary insight into the nature of the units activated by different prompt types, suggesting that only natural language prompts recruit a genuinely linguistic circuit.

## 1 Introduction

Neural language models (LMs) are parameterized probabilistic models that can assign a probability to any sequence of language tokens. Given that they are trained on huge amounts of natural language, we expect their statistics to mimic those of the latter. In this paper, we study what happens when a LM trained on English must process "unnatural language", that is, sequences that are extremely unlikely in English, as they are syntactically and semantically ill-formed.

We tackle this topic through the lens of *machine-generated prompts*, that is, automatically discovered input token sequences that optimize the model's performance in a target zero-shot task (Shin et al., 2020; Deng et al., 2022). It has indeed

been widely observed that such prompts, while empirically effective, consist of nonsensical sequences of jumbled tokens. For example, using the popular AutoPrompt algorithm of Shin et al. (2020) and the OPT-1.3b language model (Zhang et al., 2022), we found that the prompt "*[X] Antarctica = sequelsStationrough [Y]*" outperforms reasonable human-crafted prompts such as "*[X] belongs to the continent of [Y]*" on the task of retrieving the continent a geographic body belongs to. Even more extremely, recent prompt generation methods find sequences of embedding vectors that do not correspond to items in the model vocabulary, but still outperform both human-crafted and machine-derived discrete prompts (Lester et al., 2021; Liu et al., 2023; Zhong et al., 2021). This state of affairs is paradoxical: why does a LM that has been trained to reproduce the statistics of natural language respond better to input sequences that are completely outside this distribution?

We present a detailed comparative study of how LMs internally process manually-crafted prompts and both discrete and continuous machine-generated prompts. While we do not solve the puzzle of why linguistically ill-formed machine-generated prompts are better than human prompts, we discover that there are fairly deep differences characterizing the various prompt types through all the processing stages of a LM, suggesting that the latter has fortuitously developed a distinct pathway to process unnatural language.

## 2 Related work

**Understanding prompts.** The advent of zero-shot prompting stimulated interest in the linguistic and semantic properties of prompts.[1] For example, Webson and Pavlick (2022) showed

---

[1]There is also related work on the effect of ablations such as word order permutations in the context of models fine-tuned for a specific task, such as natural language inference (e.g., Gupta et al., 2021; Pham et al., 2021; Sinha et al., 2021a,b).

that, with minimal fine-tuning, highly semantically irrelevant prompts can be as effective as prompts with pertinent semantic content. Starting with Wallace et al. (2019) and Shin et al. (2020), the fact that inscrutable machine-generated discrete prompts outperform natural language sequences has also attracted attention. For example, Deng et al. (2022) showed that constraining machine-generated prompts to be more "language-like" harms performance. Ishibashi et al. (2023) and Rakotonirina et al. (2023) studied how various ablations affect the performance of machine-generated prompts. The second study also demonstrated that it is possible to find discrete machine-generated prompts that are effective across a range of LMs. Khashabi et al. (2022) found that continuous prompts can be optimized to be near *any* arbitrary text in embedding space, while being equally effective. These studies focus on properties of the prompts themselves. We complement them with an analysis of how LMs respond when exposed to these prompts.

**Understanding LMs** More generally, understanding how LMs process unnatural linguistic input contributes to our understanding of their inner workings. Therefore, our study is also related to work on *interpretability* (Lipton, 2018), defined as the analysis of a trained model's decision policy. In particular, one can approach neural network interpretability by adopting a *mechanistic* paradigm, consisting in directly studying the weights and their activation in order to reverse-engineer the neural network. Successful mechanistic insights have been obtained in computer vision (Voss et al., 2021; Olah et al., 2020). Cammarata et al. (2020) is an example of mechanistic interpretability applied to Tranformer LMs. In this context, the Transformer feed-forward layers have been shown to behave like key-value memories (Geva et al., 2021). Notably, as shown in Dai et al. (2022), these memory slots, also called *knowledge neurons*, encode specific concepts acquired during pre-training. Even more interesting, manually editing these memories allows to causally control the prediction output (Meng et al., 2022), suggesting that they play a central role in language processing (see also Geva et al., 2022). In the present paper, we show that unnatural language processing is achieved by recruiting different knowledge neurons than the ones used for natural language processing.

# 3 Setup

## 3.1 Language model and tasks

**OPT family LMs** We conduct our analyses on OPT-350m and OPT-1.3b (Zhang et al., 2022), two pre-trained auto-regressive Transformer-based models trained on The Pile corpus (Gao et al., 2020), whose pre-trained weights are publicly available from HuggingFace. We choose auto-regressive models since LM development has increasingly shifted to this class, and OPT models since, in informal experiments, we found them to perform better on our tasks than comparable auto-regressive models available from Hugging-Face (e.g., the GTP2 family). OPT models use a vocabulary set composed of 50,265 items.

**Knowledge-retrieval tasks** We base our experiment on the LAMA dataset (Petroni et al., 2019). Initially designed to probe factual knowledge and commonsense in LM, this dataset is a collection of ⟨r, s, o⟩ triplets describing a relation $r$ between a subject $s$ and an object $o$, e.g.: ⟨*continent of*, Lavoisier Island, Antarctica⟩. In particular, we use the TREx (ElSahar et al., 2018) subset, whose test set contains 41 relations, each with up to 1,000 tuples. All the machine-generated prompts are trained using the data collected by Shin et al. (2020), also containing 1,000 tuples per relation. Each relation defines a different knowledge retrieval task. We focused on these tasks because they require semantically contentful prompts (e.g., for the relation above, the prompt must carry some geographic information), as opposed to other setups where a prompt might simply have to describe the task at a meta-linguistic level ("*translate the following sentence into Chinese*"; "*does paragraph X entail paragraph Y?*", etc.).

## 3.2 Prompts

**Terminology** We refer to different methods to derive prompts as *prompt types*. We refer to the actual token sequences generated by a method for a certain task as *templates*.

**Human prompts** Human prompts (*human*) come from an augmented version of PARAREL (Elazar et al., 2021). PARAREL provides a set of near-paraphrase templates capturing each LAMA relation, e.g. "*[X] belongs to the continent of [Y]*". ParaRel enlarged the initial templates provided by LAMA using paraphrases from LPAQA (Jiang et al., 2020) and additional patterns mined from

Wikipedia. Each prompt was then evaluated by a set of human experts. We further manually augmented the set with more paraphrases, and we cleaned the prompt set, e.g., by removing templates not adapted to auto-regressive LMs.

**Machine-generated prompts**  We compare *human* prompts with both discrete (*M-disc*) and continuous (*M-cont*) machine-generated prompts. The discrete ones are obtained using the popular Autoprompt (Shin et al., 2020) algorithm. For a given task, this algorithm generates a sequence of $N$ tokens relying on a gradient-guided search in the discrete LM's vocabulary space. We set template length to $N = 5$, as it is the average human prompt length. The continuous machine generated prompts are obtained using Optiprompt (Zhong et al., 2021). For each task, Optiprompt generates a sequence of $N$ continuous vectors through optimization in the LM's embedding space. Similarly to Autoprompt, we set $N = 5$. Machine-generated prompts are extracted using the LAMA-TREx training set (see above). 10 templates are obtained for each task by initializing training with different random seeds.

**Template filtering**  We only use tasks for which we have, for each prompt type, at least one template reaching $> 10\%$ accuracy. We end up with 5.9 *human*, 8.3 *M-disc*, and 9.0 *M-cont* templates on average per task (across 21 tasks) for OPT-350m, and 6.3 *human*, 8.9 *M-disc*, and 10 *M-cont* templates on average per task (across 24 tasks) for OPT-1.3b.[2]

### 3.3  Diagnostic metrics

**Accuracy**  We measure the effectiveness of a prompt type (*human*, *M-disc* or *M-cont*) by computing its micro-accuracy (following Zhong et al. (2021)), defined as the proportion of cases where the prompted LM succesfully assigned maximum completion probability to the ground-truth object. We average across templates and LAMA tasks. It is worth noting that, contrary to other works, we did not perform any filtering on the LM's output.

**Input perplexity and output entropy**  We measure the average perplexity for each prompt type, defined as the exponentiated average negative log-likelihood of a "[subject] [template]" sequence, averaged across subjects, relations and templates. To characterize the LM probability distribution output, we also measure the average Shannon

---
[2]We attach the filtered template list as a supplementary archive.

entropy of the output probability vector computed across all samples of the evaluation set.

**Attention distribution**  We quantify how attention is distributed over input tokens following Ramsauer et al. (2021). For each attention head of each layer, we compute the average minimal number of attention values required to get a cumulative softmax probability mass of 0.90. This value ranges from 0% to 100%. Intuitively, given a row of an attention map of a transformer layer, it corresponds to the number (in %) of attention values you have to sum to reach 90% of the total attention. Because attention values are normalized, if the attention is flat then the score will be 90%. In contrast, if all the attention is focused on one token, then the score will be close to 1. This score decreases as the attention distribution becomes more peaky.

**Knowledge neuron activation overlap**  Motivated by Geva et al. (2021) and Dai et al. (2022), who empirically demonstrated that Transformer feed-forward (FF) layers act as key-value memories, or *knowledge neurons*, we measure the activation overlap of the intermediate FF units (corresponding to memory keys) between different prompt types. More formally, for a given Transformer layer, let $x \in \mathbb{R}^d$ be the token-wise hidden representation contextualised by the self-attention operation. The FF layer can be expressed as:

$$u = f(x \cdot K^T + b_K) \tag{1}$$

$$FF(x) = u \cdot V + b_V \tag{2}$$

where $V, K \in \mathbb{R}^{d \times d_m}$ are the FF parameters, $b_K, b_V$ their respective biases, and $f(\cdot)$ the non-linearity. $K$ can be seen as a set of $d_m$ keys, giving access to $d_m$ "memory slots" stored in $V$. In order to quantify which knowledge neurons are being accessed during prompt processing, we look at the units $u \in \mathbb{R}^{d_m}$ corresponding to the weights associated to each key-value pair (Eq. 1). A high overlap means that the prompts activate the same knowledge neurons, indicating similar processing by the LM. On the opposite, a low overlap suggests that the prompts trigger different activation pathways in the LM. The measure is described in more detail in Appendix A.

**Input similarity**  For each pair of same-task templates, we measure the cosine similarity of their embedded representations. We then compute the average to get similarities at the prompt-type level.

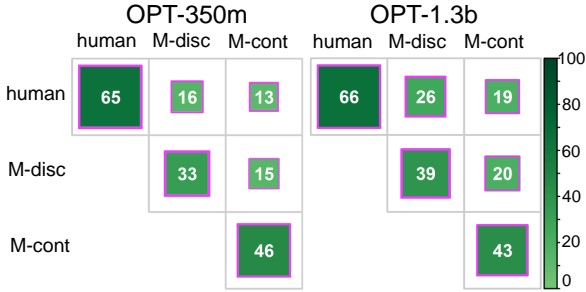

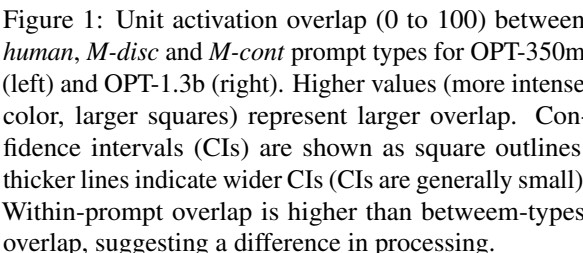

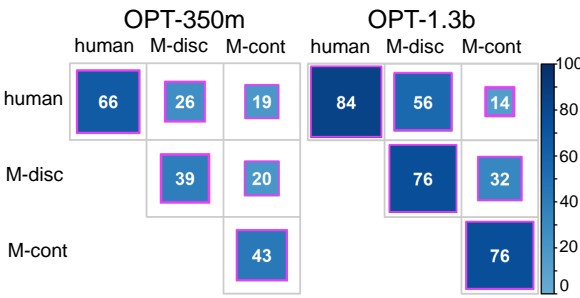

Figure 1: Unit activation overlap (0 to 100) between *human*, *M-disc* and *M-cont* prompt types for OPT-350m (left) and OPT-1.3b (right). Higher values (more intense color, larger squares) represent larger overlap. Confidence intervals (CIs) are shown as square outlines: thicker lines indicate wider CIs (CIs are generally small). Within-prompt overlap is higher than betweem-types overlap, suggesting a difference in processing.

Figure 2: Percentage input similarity between *human*, *M-disc* and *M-cont* prompt types for OPT-350m (left) and OPT-1.3b (right). Higher values (more intense color, larger squares) cue high similarity. Within-prompt-type similarities (the scores on the diagonal) are generally higher than similarity between types. Note that the absolute values of the input similarity obtained with both model sizes are not directly comparable due to a difference in the input dimension (2048 *vs.* 1024).

**Output agreement** For each pair of same-task templates, we measure the proportion of test cases where the templates lead to the same prediction. We then compute prompt-type-level averages.

**Uncertainty quantification** We provide the uncertainty estimation of our measurements by computing the 95% Confidence Interval (CI) of each measure. In Table 1, the CI associated to each metric (accuracy, perplexity, attention distribution score and output entropy) is obtained by computing the 0.025 and 0.975 quantiles given the list of scores obtained with each templates of a given prompt type (note that each template's score is averaged at the level of the relation). 95%CI in Figure 1,2,3, and Table 2 are obtained using bootstrapping by randomly sampling with replacement from the list of templates (the number of resamples is found by incrementally increasing it until the uncertainty estimation converge).

## 4   Processing machine-generated prompts

We experimentally demonstrate that differences between human and machine-generated prompts exist at three different levels: (1) at the input level, when comparing prompt types in the embedding space, (2) at the output level, when analyzing predictions and output probabilities, and (3) at the level of intermediate activation, indicating a difference in processing at work in the LM. We conclude this quantitative analysis by showing that, although these metrics are correlated when compared within the same prompt type, the correlation is weak between prompts of different types, leading to a number of

counterintuitive patterns in LM prompt processing.

### 4.1   Human and machine-generated prompts are processed differently.

**High accuracy and high perplexity** As confirmed in Table 1, the main motivation to use machine-generated prompts is their good performance, *M-cont* prompts outperforming *human* ones by +25pts. This higher accuracy comes along with lower output entropy, suggesting better LM calibration, where a larger mass of the output probability distribution is concentrated on the correct token.[3] However, prompt perplexity – quantifying the degree of predictability of a token sequence given an LM – is two order of magnitude higher for *M-disc* than for *human* templates.[4] We discuss this further in Section 4.2 below.

**Low activation overlap of knowledge neurons** Activation overlap statistics are provided in Figure 1. For both OPT-350m and OPT-1.3b, we observe that, while within-prompt-type overlap is mild or high, ranging from 33 to 66 (on a 0-to-100 scale), between-prompt-type overlap is always

---

[3]Calibration in LM analysis (e.g., Liang et al., 2023) refers to the confidence that a model has in its predictions when the latter are correct. Our output entropy measure does not directly correlate confidence and accuracy. However, as machine prompts are in general more likely to trigger the correct output and, at the same time, they have lower output entropy, the global trends do suggest that they tend to produce correct answers with more confidence. We informally use the term "calibration" to refer to this property.

[4]Due to their continuous nature, there is not trivial way to estimate perplexity for *M-cont* prompts.

| | OPT-350m | | | OPT-1.3b | | |
|---|---|---|---|---|---|---|
| | *human* | *M-disc* | *M-cont* | *human* | *M-disc* | *M-cont* |
| Accuracy | 29.5 | 43.4 | 54.9 | 28.8 | 46.1 | 58.0 |
| [95% CI] | [11.5, 65.0] | [17.0, 79.5] | [20.7, 86.0] | [10.4, 78.2] | [15.1, 83.4] | [23.8, 89.6] |
| Perplexity ($10^3$) | 0.60 | 40.9 | - | 0.40 | 30.3 | - |
| [95% CI] | [0.1, 1.9] | [16.0, 95.0] | | [0.04, 1.48] | [2.0, 911.3] | |
| Attention distribution (%) | 34.4 | 30.0 | 23.2 | 30.8 | 28.7 | 29.4 |
| [95% CI] | [29.2, 39.7] | [27.3, 32.5] | [21.1, 25.5] | [17.0, 85.8] | [16.6, 84.4] | [14.3, 74.7] |
| Output entropy | 5.00 | 4.30 | 2.10 | 4.70 | 3.90 | 2.10 |
| [95% CI] | [3.2, 6.0] | [1.9, 5.7] | [0.5, 4.3] | [1.7, 6.0] | [1.3, 6.4] | [0.4, 5.9] |

Table 1: Human and machine-generated prompts (both *M-disc* and *M-cont*) significantly differ in at least four aspects: (1) machine-generated prompts outperform the human ones in terms of accuracy; (2) they are also better calibrated on average, given their lower output entropy; while at the same time (3) machine-generated prompts are less predictable by the LMs, reaching significantly higher perplexity; (4) in machine-generated prompts, attention is concentrated on a smaller amount of tokens. For technical convenience, perplexity is not computed for *M-cont*.

low, ranging from 13 to 26. This pattern is more pronounced when comparing *human* and *M-cont*. Between-prompt overlap tends to be higher with OPT-1.3b, suggesting that larger LMs could show a convergence of human and machine-generated prompts (this remains to be further explored). The low-overlap result is confirmed by the diagnostic classifier analysis presented in App. B, that shows that a simple linear classifier can distinguish between any prompt type pair based on activation patterns on any layer of either LM.

**Attention is focused on fewer tokens** As transformer behaviour is a by-product of both FF and attention layers, we also look at the difference in attention distributions, shown in Table 1. Here again, we observe a clear distinction between human and machine-generated prompts, the latter leading to attention being focused on a smaller amount of tokens. Recall that prompt length is a hyperparameter of automated prompt induction algorithms, fixed at 5 tokens without tuning. This result might suggest that the algorithms only associated meaningful information to a subset of these tokens.

**Machine prompts are drifting away from Human prompts in the input space.** Figure 2 shows that, for both OPT-350m and OPT-1.3b, input similarity within prompt types is higher than similarity between different prompt types. In particular, the input similarity between *human* and Machine prompts dramatically decreases when moving from discrete to continuous prompts.

### 4.2 Surprising aspects of LM processing

The significant differences that emerged between human and machine prompt processing suggest that these prompt types trigger different decision pathways. Furthermore, they provide interesting insights concerning the nature of LM processing, and, in particular, how it can occasionally be quite counter-intuitive. We explore this by considering some unexpected correlation patterns.

**Perplexity does not predict accuracy** Gonen et al. (2022) reported a negative correlation between perplexity and effectiveness of handcrafted prompts. However, we observe that, when using machine-generated prompt, it is possible to reach a higher prediction accuracy while having a higher perplexity. Thus, counter-intuitively, perplexity does not necessarily predict effectiveness.

**Input similarity does not predict output agreement** The *Input predicts Output?* column of Table 2 measures the correlation between embedding-space similarities of same-task templates (e.g., a *human* and a *M-disc* template for the *continent of* relation) and the rate of output agreement (defined as the portion of times different templates lead to the same prediction) for the corresponding templates. There is a significantly lower correlation when templates belonging to different prompt types are compared, especially when comparing human vs. machine-generated templates (e.g., *human* vs. *M-disc* templates), than for within-type comparisons (e.g., different *M-disc* templates for the same task). When comparing different prompt types, counter-intuitively, the degree of similarity

| | Input predicts Output? | Input predicts Activation? | Activation predicts Output? |
|---|---|---|---|
| | OPT-350m | | |
| *human* vs. *M-disc* | **0.11** [-0.06, 0.27] | **0.21** [0.16, 0.26] | 0.01 [-0.05, 0.07] |
| *human* vs. *M-cont* | **0.14** [0.01, 0.26] | **0.06** [0.02, 0.09] | **-0.04** [-0.08, -0.00] |
| *M-disc* vs. *M-cont* | **0.30** [0.18, 0.42] | **0.06** [0.03, 0.08] | **0.02** [-0.01, 0.05] |
| *human* vs. *human* | **0.53** [0.43, 0.62] | **0.73** [0.69, 0.77] | **0.66** [0.59, 0.72] |
| *M-disc* vs. *M-disc* | **0.54** [0.45, 0.63] | **0.85** [0.83, 0.88] | **0.55** [0.49, 0.63] |
| *M-cont* vs. *M-cont* | **0.63** [0.55, 0.72] | **0.74** [0.69, 0.78] | **0.54** [0.47, 0.61] |
| | OPT-1.3b | | |
| *human* vs. *M-disc* | **0.18** [0.07, 0.28] | **0.24** [0.18, 0.30] | **0.13** [0.05, 0.22] |
| *human* vs. *M-cont* | -0.06 [-0.21, 0.08] | **0.08** [0.03, 0.14] | -0.03 [-0.08, 0.03] |
| *M-disc* vs. *M-cont* | **0.12** [0.01, 0.22] | **0.03** [-0.01, 0.07] | -0.00 [-0.05, 0.04] |
| *human* vs. *human* | **0.58** [0.52, 0.66] | **0.65** [0.61, 0.68] | **0.60** [0.55, 0.65] |
| *M-disc* vs. *M-disc* | **0.64** [0.58, 0.70] | **0.89** [0.87, 0.90] | **0.61** [0.55, 0.67] |
| *M-cont* vs. *M-cont* | **0.78** [0.73, 0.83] | **0.74** [0.71, 0.77] | **0.53** [0.49, 0.57] |
| *Notation*: **average** [95% confidence interval] | | | |

Table 2: Pearson correlations between input similarity, output agreement and activation overlap. First, we compute a single comparative statistic (input similarity, output agreement or activation overlap) for each pair of prompts in some comparison set (e.g., *human* vs. *M-disc* or *human* vs. *human*); then, for each comparison set, we look at the correlation across prompt pairs between two statistics (e.g., input similarity vs. activation overlap). Within-type correlations range from mild to high. Between-type correlations are significantly lower. These low correlations highlight counteractive aspects of LM language processing. Results in **bold** are significant (p < 0.01). We provide the average and [95%CI interval] correlations obtained using bootstrapped uncertainty estimation.

is not a good predictor of whether the templates will trigger the same output or not.

**Activation overlap is only weakly correlated with output agreement and input similarity** Table 2 also provides correlations between activation overlap and input similarity (*Input predicts Activation?*) or output agreement (*Activation predicts Output?*) across different pairwise prompt type combinations. Within a prompt type, these correlations are mild or high, with the higher correlations pertaining to input similarity. On the contrary, activation overlap ceases to be correlated to either input similarity or output agreement as soon as we compare different prompt types. This drop in correlation highlights the complexity of LM internal processing. Without any prior on input type, it is difficult to predict the decision pathway that will be used by the model, even in the presence of high input or output similarities.

## 5 A closer look at the typical units of each prompt type

### 5.1 Unit distribution across layers

The low activation overlap between prompt types reported in Section 4 taught us that machine-

generated prompts trigger units which are distinct from the ones triggered by human prompts. The units that are most often activated by the various prompt types also appear, to some degree, to be distributed differently across layers (cf. Figure 3). In particular, machine prompts display a tendency to activate more units on the last layer and, especially, on the first one (it is worth recalling that this is the first proper Transformer layer, and not the embedding layer). The *M-disc* profile lies somewhere between *human* and *M-cont*, confirming the trends already observed in Section 4.

### 5.2 Profiling prompt-type-typical units through associated vocabulary items

Having shown that the three prompt types activate different pathways through the network, we seek now some insights into the nature of the units characterizing these different pathways.

**Methodology** We identify those units that are both *typical* of a single prompt type across tasks, and significantly impacting the network output distribution, in the sense that their gradients w.r.t to the max output probability are in the top quartile of all network units (recall that, as usual, we focus on those units we identified as knowledge neu-

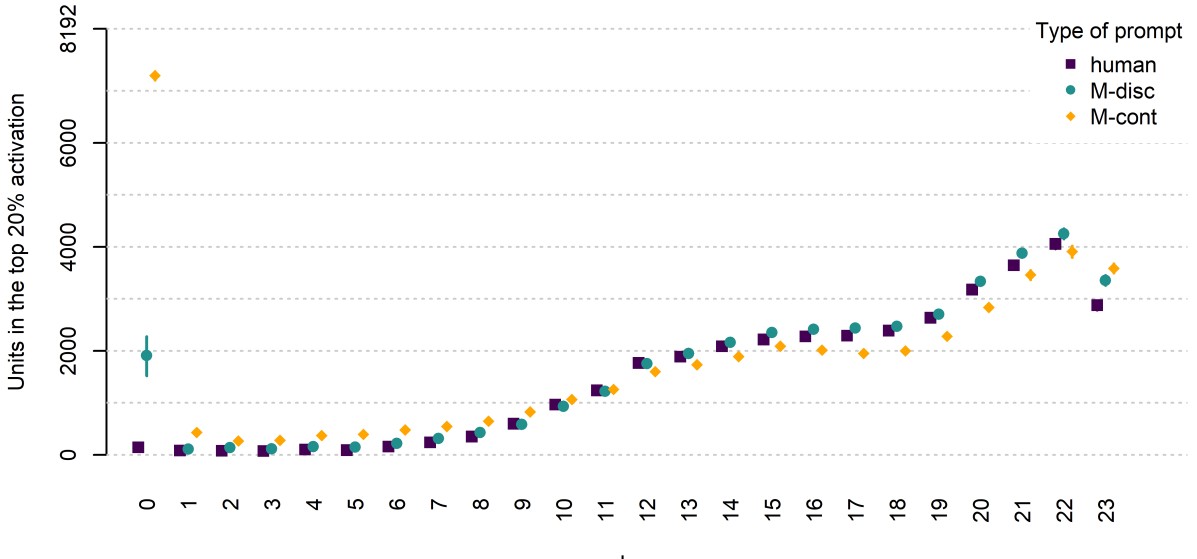

Figure 3: For each prompt type, we plot the number of units belonging to the top 20% most activated units (overall across prompt types). *M-cont* and *M-disc* have significantly more highly activated units than *human* on the first layer, with the effect particularly strong for *M-cont*. There is also a weaker tendency for the machine prompts to activate more units on the last layer compared to the *human* ones. Data from OPT-1.3b.

rons). We define the typical units for prompt A as those that are among the top 20% most frequently activated by this prompt type, while at the same time being among the bottom 20% least frequently activated by prompt types B and C.[5] This filtering procedure leaves 14 *human* (resp. 6), 4 *M-disc* (resp. 4) and 58 *M-cont* (resp. 238) units for OPT-350m (resp. OPT-1.3b). As a sanity check, we also repeated the analysis with laxer thresholds involving more units, and the results were similar to the ones we report here.

Next, we associate each selected unit to a set of items from the LM vocabulary that strongly trigger its activation. Using the Wikipedia corpus,[6] for each item in the vocabulary we save the average unit activation in a forward pass. We sort the resulting matrix to get, for each unit, the top 500 vocabulary items leading to the strongest activation. We extract both input items, recording unit activation when an item is in the input sequence, and output items, recording activation when an item is predicted by the LM. We apply lower-casing and initial-space stripping on the resulting vocabulary set. More details are provided in Appendix C. This method has been chosen for its simplicity. However, it is also noisy and sensitive to rare but "exciting" tokens (e.g., *magikarp*, see Appendix C Ta-

ble 4). Improving unit-item association extraction is left to future work.

Having obtained the list of vocabulary items associated to each unit, we count how many times each vocabulary item occurrs in association with any typical unit of each prompt type (as defined at the beginning of this Methodology paragraph), obtaining 3 frequency lists, one for each prompt type. We compare the relative frequencies of each vocabulary item in each list to determine which vocabulary items are most distinctively associated to (the set of typical units of) each type. In particular, using a standard method from corpus linguistics, we compute the *local Mutual Information* score (Evert, 2005) between each vocabulary item $v$ and each prompt type $t$:

$$\text{LMI} = |v,t| \log \frac{P(v,t)}{P(v)P(t)}$$

where $|v,t|$ counts the occurrences of $v$ in the $t$ list, the joint probability $P(v,t)$ is estimated based on $|v,t|$; $P(v)$ is estimated using the cumulative occurrence count of $v$ in all lists, and $P(t)$ is the total number of occurrences of any item in the $t$ list. Table 3 reports the top-30 *input* vocabulary items ranked by LMI for each prompt type and both LMs.

**Machine prompts recruit "non-linguistic" units**
Looking at the OPT-350m results first, nearly all characteristic *human* items are well-formed words,

---

[5]A unit is activated when its value is greater than 0.
[6]Subset "20220301.en" from HuggingFace

| *human* | | | *M-disc* | | | *M-cont* | | |
|---|---|---|---|---|---|---|---|---|
| | | | | *OPT-350m* | | | | |
| whats | gazed | ful | handler | (& | 361 | ÔøΩ | stat | //// |
| name | nifty | darn | expr | avascript | ancel | ÔøΩÔøΩ | page | pwr |
| why | devs | freaking | iterator | cpp | yout | }); | {\ | sts |
| fuck | much | these | terness | addons | risome | ()); | 0000 | ]} |
| noticed | like | have | hillary | \- | frameworks | .......... | += | table |
| really | daddy | wanna | filename | lication | ithub | crossref | }; | interstit. |
| thats | likes | what | easy | 702 | ÔøΩ | println | stats | /** |
| does | honestly | crappy | disabled | 502 | errors | warn | )); | ({ |
| thing | workaround | relent | rc | 601 | poons | }) | crip | ///////// |
| goddamn | bothers | been | json | sacrific | inline | ——— | – | debug |
| | | | | *OPT-1.3b* | | | | |
| undermines | severe | conducive | â© | äĥ® | attle | âķĝ | leilan | looph |
| curls | dictates | frowned | appalling | extreme | cram | âłĝâłĝâł | âłâłâłâł | ^^^^ |
| makes | tempt | optimized | mal | eff | dop | endif | everal | âκĵ |
| will | unfold | remain | early | monitor | egregious | canaver | archdemon | xff |
| does | bounces | reap | complex | schizophren | rep | citiz | marketable | ../ |
| manifests | persist | stroll | hou | rece | pass | %%%% | // | 0000 |
| prevail | outweigh | shines | gres | fail | insanely | ó | âħ¢: | aeper |
| haunt | haun | hangs | crazy | kinda | ãĥ¨ | âķ | dilig | nanto |
| meshes | smokes | governs | delay | shitty | prototype | %% | ............... | cryst |
| wipes | poised | fills | capital | /// | devices | ##### | â·â· | leban |

Table 3: Top 30 vocabulary items associated to each prompt type ranked by LMI. Machine-generated prompts respond to less language-like items than those triggered by human prompts. Nearly all *human* items are well-formed words. Many *M-disc* items, on the other hand, are non-English diacritics, special symbols or code-related terms. *M-cont* items are entirely "non-linguistic". Some strings have been abbreviated to fit column width.

and include a high number of forms cuing syntactic processing, such as function elements (*whats, why, does...*), inflected verbs (*noticed, gazed, liked...*) and modifiers (*really, much, honestly...*). A remarkable amount of *M-disc* items are coding-related terms (*handler, expr, iterator...*). Numbers and punctuation sequences that could be coding-related or web-page boilerplate (*(&, \-*) also appear, as well as a few regular words or word fragments (*Hillary, easy, sacrific...*). Finally, for *M-cont*, the items are entirely "non-linguistic", being composed of sequences of non-Latin characters or punctuation marks, as well as code fragments.

Concerning OPT-1.3b, we observe pretty much the same patterns for *human* and *M-cont*. For *M-disc*, on the other hand, together with a number of non-English diacritics and special symbols, there is a strong increase in regular words and word fragments, although the latter still clearly differ from those associated to *human* prompts, in that syntax-related items, such as function words or inflected verb forms, are largely missing. In line with what we observed in Figure 1, we thus observe a cline on which, at least for *M-disc*, the difference in processing human and machine-generating prompts decreases with model size.

We tentatively conclude that machine prompts are not only triggering different activation path-ways, but that the units involved in these pathways tend to respond to less language-like items than those triggered by human prompts. Note that these units are spread across the layers of the network, so that we are not only recording low-level differences in processing the input strings or vectors.[7] Moreover, the results are largely mirrored by those obtained when associating units with output instead of input vocabulary items (Table 5 in App. D).

Recall that our analysis is based on units that are not only highly typical of a prompt type across relations, but also in the top gradient quartile, suggesting that they significantly contribute to the model's output distribution. It is puzzling that units that mostly respond to coding fragments or unusual characters could lead the network to produce the correct next token in the semantic tasks we are studying. We conjecture that distributed activation from such units might nudge the network towards the right output semantic fields through connectiv-

---

[7]The selected typical *human* units occur in layers 4th to last of OPT-350m and layers 2 to 14 of OPT-1.3b (counting from 0). The *M-disc* units range from layers 3 to 10 of OPT-350m and 2 to 13 of OPT-1.3b. The *M-cont* units are the only ones where, as expected given the distribution illustrated in Figure 3, a significant proportion occurs on the first (0-th) layer (about one third for OPT-350m and one fourth for OPT-1.3b), but the remaining two thirds/three fourths range from layers 2 to 21 and 1 to 10, respectively.

ity pathways that fortuitously arose during network training. This is an important topic for future work.

# 6 Discussion

We have studied the phenomenon of linguistically and semantically opaque machine-generated prompts from the perspective of how LMs process them, compared to human-crafted ones. Our study has important **Limitations**, that are discussed in the relevant section below. However, at least for the prompt generation methods, LMs and tasks we explored, we can draw some general conclusions.

**More than a "happy accident"** Our evidence suggests that the differences between human and machine-generated prompts are not just superficial, but affect all levels of network processing, and result in the activations of qualitatively different units. Some of these units are stable across semantic tasks, suggesting that they are more generally recruited to process any "unnatural" input. Moreover, contrary to what one could reasonably predict, there is some evidence that machine prompts are more robust than human ones, in the sense that they achieve better output calibration.

It's unlikely that the LM has been exposed to anything like *M-disc* prompts during its initial training, and definitely it could not have seen out-of-vocabulary *M-cont* prompts, so we can only assume that the special pathways triggered by these prompts arose through unforeseen side effects of pre-training.[8] However, they seem to be more than just lucky connectivity accidents exploited by specific prompts to solve specific tasks, or else it would be difficult to explain the overall low entropy of machine prompt predictions and the commonalities in the units they activate. Moreover, there is evidence that *M-disc* prompts can transfer across Transformer-based LMs (Rakotonirina et al., 2023; Zou et al., 2023), suggesting that unnatural language pathways might arise from the interaction of general characteristics of the Transformer architecture with Web-derived training data that are partially shared across many current pre-trained LMs. We must defer a better understanding of the nature of these unnatural pathways to future work.

In particular, we plan to zoom further in into the processing of specific templates, tracking their processing throughout the network with methods such as the vocabulary-based unit analysis of Section 5.

**On investigating unnatural language** We believe that investigating "unnatural language" as we did here (see also Khashabi et al., 2022; Ishibashi et al., 2023; Rakotonirina et al., 2023) should be a central concern to NLP for at least three reasons.

First, *understanding* why LMs work as well as they do, and what are their failure modes, is one of the questions with the broadest scientific and societal implications we can ask today. It would however be dangerously limiting to narrow our investigation to how LMs process *natural* language only, ignoring their behaviour when presented inputs outside their training distribution.

Second, unnatural language can be exploited for negative purposes, as shown by Wallace et al. (2019) and Zou et al. (2023), who derived apparently nonsensical prompts that could steer multiple LMs' responses towards harmful behaviour, such as generating racist language.

Finally, there is recent interest in letting LMs directly communicate with each other to jointly solve tasks or to build a community (Park et al., 2023; Zeng et al., 2023). Based on our evidence, it might be pointless to insist that LM-to-LM communication takes place in natural language, given that LMs might share information more efficiently through unnatural prompts. Conversely, if being able to decode the communication flow is important (e.g., for safety reasons), care must be taken to stop LMs from drifting into unnatural language.

For all these reasons, we hope that our preliminary contribution will encourage our community to pay more attention to the phenomenon of unnatural language processing.

---

[8] We experimentally verified that model training is necessary for effective unnatural prompts to arise. We ran both Autoprompt and Optiprompt on 3 distinct random initializations of OPT-1.3b with the same hyperparameters as in our main experiments, and found that the resulting *M-disc* prompts achieved 0% accuracy in nearly all cases, whereas *M-cont* where at best able to retrieve the majority output of a task.

## Limitations

- **Main limitation:** We presented an extended study of *how* two pre-trained language models process human and machine-generated inputs, but we did not provide an account of *why* we are observing the processing differences we are seeing. We noticed, for example, that *M-cont* prompts activate units associated to punctuation marks and special characters. We do not know, however, in which way these units contribute to retrieving the correct answer in the target semantic tasks, nor how the optimization procedure chances upon them. This is our priority for future work.

- Our work is limited to the OPT family of models trained on the English language, to the LAMA semantic tasks and to the AutoPrompt and OptiPrompt prompt extraction methods. A straightforward direction for future work is to extend our analysis to more models (including instruction-tuned models, as instruction tuning might have a significant impact on how models respond to unnatural input), languages, data-sets and prompt extraction algorithms.

## Ethics Statement

The advent of publicly accessible LM interfaces such as ChatGPT has heated up the debate around the broader impact of LMs. While there is a variety of possible societal issues to consider (Weidinger et al., 2022), we believe that a better understanding of how LMs process information is a crucial part of bias and harm containment. If we do not understand the models, we cannot control their behaviour, and we are exposed to intentional adversarial attacks and other forms of unintentional model misuse. The very existence of completely opaque but empirically effective machine-generated prompts is proof of how counterintuitive the behaviour of LMs can be, and of how little we understand them. We thus believe that our investigations of "unnatural language processing" fit well into the broader program of improving our scientific understanding of LMs, in order to make them more predictable, controllable and, ultimately, safer.

## Acknowledgements

We thank Emmanuel Chemla, Emily Cheng, Nathanaël Carraz Rakotonirina, Xavier Suau, the members of the UPF COLT lab, the members of the Barcelona Apple Machine Learning Research group and the participants in the EviL seminar for helpful feedback and suggestions. Our work was funded by the European Research Council (ERC) under the European Union's Horizon 2020 research and innovation programme (grant agreement No. 101019291). This paper reflects the authors' view only, and the ERC is not responsible for any use that may be made of the information it contains.

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

## A Measuring the overlap of activated knowledge neurons

In Section 4, we measure and compare which knowledge neurons, that is, units found in the intermediate layers of the feed-forward Transformer blocks (Dai et al., 2022; Geva et al., 2022), are activated by different prompt types. In particular, we measure the *knowledge neuron activation overlap* (abbreviated as activation overlap). The following algorithms in pseudo-code detail how this measure is obtained.

First, we construct a Boolean matrix recording which units are activated by a template. As shown in the pseudo-code below, a unit is said to be activated if its value is greater than 0 on more than $k = 20\%$ of cases when instantiated with each of the subjects associated to the template relation. In the pseudo-code, `Relations` is the set of relations available in our task set; given a relation, `Subjects` provides the list of relevant subjects and `template(s)` instantiates the template with subject s. `Model` is the LM being used.

```python
def get_act(template, relation):
    cpt = 0
    k = 0.2
    # iterate across subject
    for s in Subjects(relation):
        inpt = template(s)
        for u in Model(inpt).units:
            cpt[u] += (u>0)
    # u is activated if >0 for
    # more than k% of inputs
    act = cpt > k*len(Subjects(relation))
    return act
```

Then, for each pair of templates (in `Templates`), we compute the intersection over union of their respective activation matrices:

```python
overlap = {}
for r in Relations:
    for t_A in Templates(r):
        for t_B in Templates(r):
            act_A = get_act(t_A,r)
            act_B = get_act(t_B,r)
            i = act_A & act_B
            u = act_A | act_B
            overlap[(t_A, t_B)] = i/u
```

Finally, we average these pairwise overlaps while filtering by prompt type (e.g., only averag-

| Layer 0, unit 248 | | |
|---|---|---|
| "( | »»»» | .............. |
| [_ | .......... | ="/ |
| :::::::: | {: | ,,,, |
| Layer 10, unit 674 | | |
| antidepress | debian | frieza |
| magikarp | minecraft | xperia |
| oneplus | awakens | bitcoin |
| Layer 16, unit 2126 | | |
| filename | windows | drm |
| sshd | misunder | rm |
| folder | pkg | vm ' |
| Layer 22, unit 3617 | | |
| everal | huge | every |
| risome | some | any |
| these | crappy | nifty |

Table 4: For each intermediate unit in the OPT's feed-forward layers we extract the set of items leading to the strongest activation on the Wikipedia corpus. As an illustration, we selected four different units with distinct profiles and displayed them in this table, along with their top 9 most associated input items, for OPT-350m. We observe varying degrees of consistency and naturalness across units.

ing the activation overlap for pairs containing one *human* and one *M-cont*).

## B Diagnostic classifiers

To complement the activation-overlap-based analysis presented in Section 4.1 of the main paper, we run a set of shallow linear "diagnostic" classifiers (Giulianelli et al., 2018) of the activations generated by the models on each layer in response to inputs from each prompt type. As usual, we focus on the activation of knowledge neurons.

**Data** As in the main paper, we use all templates with LAMA accuracy $>= 10\%$, filtering out a random subset of the LAMA P176 relation *human* templates, as this relation is greatly over-represented. We are left with 21 and 24 tasks for OPT-350m and OPT-1.3b, respectively. As we are interested in units inherently distinguishing prompt types independently of lexico-semantic aspects associated to specific templates or tasks, we partition the data so that the training and test data contain disjoint tasks (and, *a fortiori*, disjoint templates). We consider 4 such partitions, each time using data from 16 (OPT-350m)/18 (OPT-1.3b) tasks for training and 5 (OPT-350m) /6 (OPT-1.3b) for testing, such that

there is no test task overlap across the partitions.[9] We instantiate each template with 10 randomly selected subjects from the corresponding LAMA lists. For each pairwise classification task, we balance the test instances by downsampling the larger class, so that chance/majority/minority accuracies ("baselines" in Figure 4) are at 50%.

**Classifier** We use a logistic regression classifier with L1 regularization (to encourage sparseness), with the L1 term coefficient fixed at $\alpha = 0.01$. We fit the classifier with stochastic gradient descent, using the Scikit-learn toolkit (Pedregosa et al., 2011). For each of the 4 training partitions, we repeat the experiment with 5 different seeds, resulting in a total of 20 runs for each layer.

**Results.** Figure 4 reports average per-layer classification accuracies for each pairwise prompt type comparison, with standard deviations across the 20 runs. In all cases, accuracy values are well above baseline level, and typically very high. We conclude that each single layer contains units whose activation is sufficiently discriminative for each prompt type to successfully train the classifiers, despite the challenging setup in which training and test tasks (and consequently templates) are completely disjoint. Interestingly, few units on each layer suffice to discriminate prompt types (the average classifier weight sparsity across all experiments is at 99.5% with 0.2% standard deviation for OPT-350m and 99.6% with 0.4% s.d. for OPT-1.3b). The easiest distinctions involve *M-cont* as one of the classes, confirming that out-of-vocabulary embeddings make continuous prompts particularly different from natural language. Indeed, it's remarkable that distinguishing *M-disc* from *M-cont* is generally easier than distinguishing between *M-disc* and *human* prompts.

To conclude, the classification experiments bring strong convergent evidence that genuinely different pathways characterize different prompt types across all layers of the network.

## C  Unit/vocabulary item association

**Implementation details** When extracting unit/vocabulary-item association, we empirically set the window size to 15. This value is reasonable close to prompt size (5 in average), while containing a sufficient amount of tokens to get a

meaningful context. In addition, we set the window stride to 15 to save computation time. Furthermore, due to our limited computation resources, and given the size of the Wikipedia corpus (6B), we only used 66% of the data for OPT-1.3b.

**Samples** As illustrated in Table 4, we came up with a large diversity of unit profiles, some of them being associated to more or less linguistically valid items, and with varying degree of semantic consistency.

## D  Profiling typical units by output vocabulary analysis

Table 3 in the main paper reports the 30 *input* vocabulary items with the largest LMI with respect to each prompt type. Table 5 here reports the top-30 *output* items. We largely confirm the same trends, although we do notice an overall tendency for the units triggered by machine prompts to be associated to more "language-like" output material, which makes sense as ultimately these prompts do produce well-formed task-relevant outputs.

[9]As we have 21 total tasks meeting our conditions for OPT-350m, one of the 4 partitions used for this model consists of 15 training and 6 test tasks

---

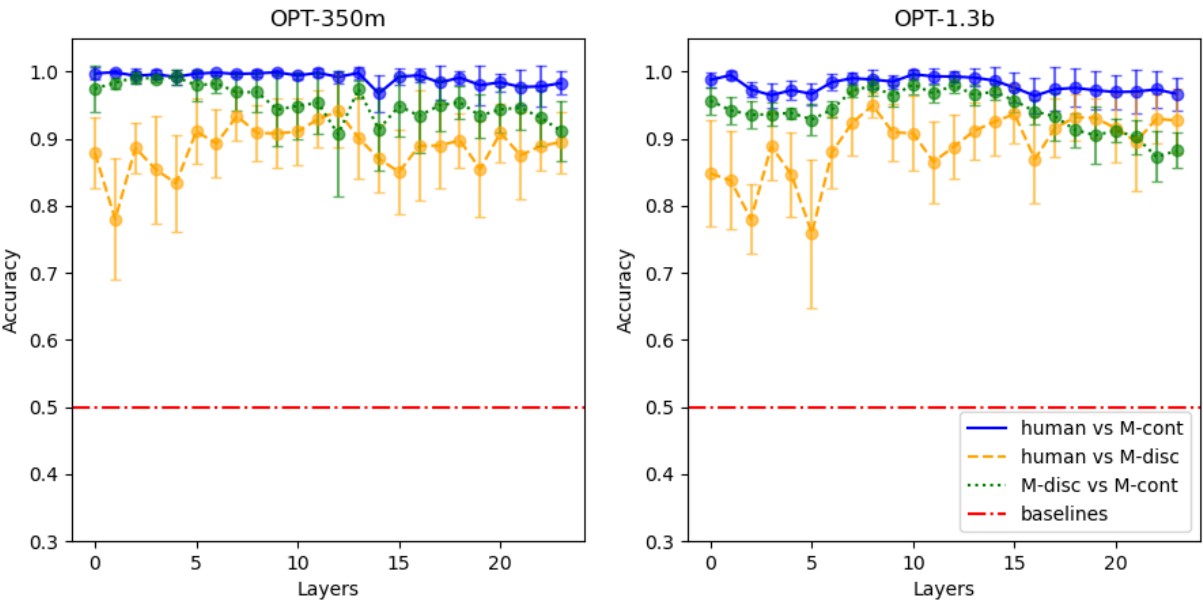

Figure 4: Average accuracies and standard deviations for 20 runs of the pairwise prompt-type classification experiments across the 24 layers of the networks.

| human | | | M-disc | | | M-cont | | |
|---|---|---|---|---|---|---|---|---|
| *OPT-350m* | | | | | | | | |
| nobody | undone | tonight | incompet | fanc | { | ÔøΩ | ,ñà | compare |
| yesterday | before | scrambled | alluded | ridic | \` | ÔøΩÔøΩÔøΩ | ", | pref |
| okay | anybody | reasonably | \\ | 782 | ``% | tracker | }); | also |
| really | plugged | messed | ): | blat | forge | ;; | moreover | comple |
| awoken | captcha | pinpoint | juxtap | xcom | physic | checking | [' | attempt |
| bother | interchangeable | right | "'\ | ":[" | tyr | supported | therefore | '); |
| corrobor | glanced | authenticated | revert | -1 | dst | meanwhile | text | 0000 |
| parsed | glean | snowball | insin | 698 | ({ | avg | ⊦ | (( |
| bothering | tweaked | earlier | faintly | invoke | 772 | insert | »> | currently |
| nailed | figured | tasted | irresist | mysql | ceremon | header | }," | prev |
| *OPT-1.3b* | | | | | | | | |
| accumulating | snug | peeled | reconc | oddly | localhost | âkǵ | â | âłǵâłǵâłǵâł |
| elic | attributable | distinctly | monstrous | vaugh | filib | âlâł | â·â | ôł |
| overpower | solely | ooz | trembling | prett | though | âli | âh ì | inst |
| substantially | unmatched | waging | check | outlandish | recip | łǁ | âłǵâł | }{ |
| fundamentally | overloaded | incred | adolesc | mpg | disag | kinnikuman | say | âiij |
| reinvest | obligatory | impecc | âī¼ | pause | acquies | %%%% | += | âli |
| unparalleled | infused | deprecated | understanding | independ | murky | === | any | ***** |
| anonym | constrained | inherently | collaps | bicy | budgetary | services | sample | i̠ı |
| overseen | sandwic | surg | disbel | scram | game | provider | âǁ | ãhi |
| ideally | achievable | tempted | unpop | foundational | billboards | âłâłâłâłâł | url | ^^^ |

Table 5: Top 30 output items associated to each prompt type ranked by LMI. Some strings have been abbreviated to fit column width.