# OpenReview forum: "Unnatural language processing: How do language models handle machine-generated prompts?"
_EMNLP/2023/Conference — EMNLP 2023 Findings_

### Official Review · Reviewer_oqtz · 2023-07-24

**Soundness:** 4

**Excitement:**

4: Strong: This paper deepens the understanding of some phenomenon or lowers the barriers to an existing research direction.

**Paper Topic And Main Contributions:**

This is an exploratory paper that attempts to characterize in detail the empirical properties of language models' (LMs) processing of human-readable vs. machine-optimized prompts. Diverse analyses suggest that machine-optimized prompts propagate through the network in qualitatively different ways than human-readable prompts that more closely resemble the models' training inputs: machine-optimized prompts have much higher perplexity than human-readable prompts, show little activation overlap with human-readable prompts, have peakier attention distributions than human-readable prompts, and have lower output entropy than human-readable prompts. Although this study is not designed to illuminate the causes of these differences, results are at least consistent with the possibility that prompt optimization works by finding and exploiting novel activation pathways through the network that work well for the target task, a finding that has also been suggested by recent closely related work (Khashabi et al 22).

I think this is a carefully executed study and a well-written paper and I support publication. Some readers may find the lack of a causal hypothesis disappointing, but I think for many this will mostly be a hindsight effect thanks to the very clear problem statement in the paper itself: the authors make a strong case that it's worthwhile to study the mechanisms by which prompt engineering works and that too few researchers are currently tackling this question. The detailed exploratory characterization of the empirical phenomenon that this paper provides will likely be an influential early step on the path toward deeper understanding of this question.

I do have some technical and presentational concerns that I hope the authors will consider addressing in revisions.

**Reasons To Accept:**

1. Clearly written
2. Detailed, well-motivated analyses
3. Potentially fundamental insights (substantial separation in the network-internal activation pathways between human-readable vs machine-optimized prompts)
4. Detailed exposition of potential weaknesses, strong grounding of this work in the ethical implications of LM interpretability

**Reasons To Reject:**

1. This work participates in a growing movement to treat LMs not only as engineering tools but as objects of scientific inquiry in their own right. As a result, questions shift from "which LM does better at X?" to "why/how do LMs do X?" This kind of work is much-needed and I strongly support it. However, I think quantitative approaches that are standard in NLP and largely used in this study (esp., leaderboard-style reporting of numerical differences between models) don't meet quantitative standards for scientific research, which place a high priority on uncertainty quantification in the form of e.g., confidence intervals, posterior distributions, hypothesis tests, etc. There are a variety of reasonable tools that could be used, but the overall goal is to distinguish the statistics of the sample from the inferences about the population, and to make explicit the reasoning by which population inferences are made from these statistics. For example, the authors say based on Table 1 that the attention distribution statistic (number of timepoints needed to cover 90% of the softmax distribution) is smaller in M-disc than human prompts. In a sense, this is uncontroversially true because the number is smaller (30 vs 34 using OPT-350m). But is this difference an inherent property of the M-disc vs human prompting strategies (that would appear again with high probability if this experiment were repeated), or is it an artifact of the particular sample of prompts used here? Without uncertainty quantification, it's hard to know whether a difference of 4 for the attention statistic big or small, meaningful or not, etc. I think this work would be more convincing if readers had a clearer idea about the distributional properties of the statistics being reported, in order to form intuitions about claimed differences.

2. The activation overlap and similarity analyses are less independent than they seem: lower cosine similarity implies lower (standardized) Pearson correlation, which in turn implies lower activation overlap (due to the lower likelihood that a unit will co-activate across conditions). In addition, the activation overlap measure introduces an experimenter degree of freedom: it requires an arbitrary frequency threshold for positive activation and selects neurons for analysis based on it in a binary manner. Why discretize a continuous space a way that seems designed to exaggerate subtle continuous differences? It seems a lot more straightforward to use simple correlation in place of the current factorization into (discrete) activation overlap and (continuous) similarity, which seem like two different views on the same data.

3. The authors make an assumption that better-calibrated models will have lower output entropy. This assumption is false in general, since without direct evaluation (i.e., of the frequency with which a prediction made with model-assigned probability *p* is correct) one cannot know whether a given model should have higher or lower output entropy. Maybe the authors have reason to believe this is true *for this specific evaluation*? If so, the reasons should be given. If not, I think they should not invoke the notion of calibration, which seems like a red herring here.

4. It's hard to figure out what is being correlated with what in Table 3. After a lot of puzzling I think I finally got it: the correlation is between two *statistics* (activation overlap, input similarity, and output agreement) computed either within or between prompt types. So the Activation-Input statistic for the human vs. M-disc condition represents the degree to which the activation overlap statistic using human promps is correlated with the input-similarity statistic using M-disc prompts (?). I'm still not sure this is right, which I think speaks to deeper presentational problems. How the numbers in Table 3 were computed and what they mean is unclear, and more explanation is needed in the text and caption.

5. I think the so-called "human" prompts are mis-labeled---to me, this label implies that human annotators wrote prompts for each training/testing instance, when in reality (if I understand the methods) these prompts were generated from the training/test instances using a hand-made codebook. I think "rule-based" or "human-readable" would be a clearer label.


**Reproducibility:**

3: Could reproduce the results with some difficulty. The settings of parameters are underspecified or subjectively determined; the training/evaluation data are not widely available.

**Reviewer Confidence:**

3: Pretty sure, but there's a chance I missed something. Although I have a good feel for this area in general, I did not carefully check the paper's details, e.g., the math, experimental design, or novelty.

---

> ### Author Rebuttal · Authors · 2023-08-28
>
> Thank you for your detailed review, with valuable comments and positive feedbacks. Please find the answers of your questions as follows:
>
> ## Statistical analyses
>
> > I think this work would be more convincing if readers had a clearer idea about the distributional properties of the statistics being reported, in order to form intuitions about claimed differences.
>
> We appreciate the thoughtful insights shared by oqtz, highlighting the importance of incorporating uncertainty quantification techniques. To address this concern, **we will be appending a 95% confidence interval (95% CI) to all the scores in the paper, achieved by considering variability across relations/tasks or bootstrapping**. It's important to emphasize that even when looking at confidence intervals obtained in this way, **the reported results maintain their statistical significance**. As an illustration, below are the updated outcomes for opt350m:
>
> ### Table 1
> We added a 95% CI with regards to the relations scores (we first averaged the sample-level scores across templates to get relation-level scores. Then we computed the final average scores and their confidence interval based on these relation-level scores).
> | Type   | Micro         	| PPL (10^3)    	| Entropy    	| Attention dist.   |
> |--------|-------------------|-------------------|----------------|-------------------|
> | Human  | 29.5 [11.5, 65.0] | 0.6 [0.1, 1.9]	| 5.0 [3.2, 6.0] | 34.4 [29.2, 39.7] |
> | M-disc | 43.3 [17.0, 79.5] | 40.9 [16.0, 95.0] | 4.3 [1.9, 5.7] | 30.0 [27.3, 32.5] |
> | M-cont | 54.9 [20.7, 86.0] | NaN           	| 2.1 [0.5, 4.3] | 23.2 [21.1, 25.5] |
>
> ### Figure 1
> We provide a bootstrap estimation of the 95% CI (we randomly sample 350 templates 1000 times):
> | Type         		 | Activation overlap     |
> |-------------------    |--------------------    |
> | Human        		 | 65 [63, 67]   		 |
> | M-disc       		 | 31 [30, 33]   		 |
> | M-cont       		 | 45 [44, 46]   		 |
> | Human vs. M-disc 	 | 16 [16, 17]   		 |
> | Human vs. M-cont 	 | 13 [13, 14]   		 |
> | M-disc vs. M-cont     | 15 [15, 15]   		 |
>
> ### Figure 2
> Same as in Fig1:
>
> | Type         		 | Input similarity     |
> |-------------------    |------------------    |
> | Human        		 | 96 [96, 96] 		 |
> | M-disc       		 | 90 [90, 90] 		 |
> | M-cont       		 | 85 [84, 86] 		 |
> | Human vs. M-disc 	 | 86 [85, 86] 		 |
> | Human vs. M-cont 	 | 48 [48, 49] 		 |
> | M-disc vs. M-cont     | 51 [50, 51] 		 |
>
> ### Table 2
> Similarly, we provide the bootstrap 95% CI of the correlations (again we randomly sample 350 templates 1000 times). We also rearrange the table to make it more clear:
>
> | Type                 		 | Input predicts Activation?     | Activation predicts Output?     | Input predicts Output?     |
> |---------------------------    |----------------------------    |-----------------------------    |------------------------    |
> | Between Human and M-disc 	 | 0.21 [0.17, 0.25]     		 | 0.01 [-0.04, 0.06]     		 | 0.11 [-0.03, 0,24]		 |
> | Between Human and M-cont 	 | 0.06 [0.03, 0,08]     		 | -0.04 [-0.07, -0.00]   		 | 0.14 [0.04, 0.24] 		 |
> | Between M-disc and M-cont     | 0.06 [0.04, 0.08]     		 | 0.02 [0.00, 0.05]      		 | 0.30 [0.19, 0.40] 		 |
> | Within Human         		 | 0.72 [0.68, 0.75]     		 | 0.65 [0.59, 0.70]      		 | 0.51 [0.43, 0.59] 		 |
> | Within M-disc        		 | 0.84 [0.82, 0.86]     		 | 0.51 [0.48, 0.59]      		 | 0.52 [0.45, 0.59] 		 |
> | Within M-cont        		 | 0.72 [0.68, 0.75]     		 | 0.51 [0.46, 0.58]      		 | 0.61 [0.54, 0.68] 		 |
>
>
> ### Figure 3
> We have incorporated the 95% confidence intervals using the same bootstrapping technique.
>
>
> ## Discretized overlap
> > Why discretize a continuous space a way that seems designed to exaggerate subtle continuous differences? It seems a lot more straightforward to use simple correlation in place of the current factorization into (discrete) activation overlap and (continuous) similarity, which seem like two different views on the same data.
>
> Thanks for the suggestion. We have devised the discretized measure because 1) we were concerned that possible patterns would have been hidden by **low-frequency noise when correlating the whole activation vectors**; 2) the discrete measure is **more straightforward to compute across layers** (so that we have a single overlap score, instead of 24); and 3) the method returns **a discrete set of “active” units that we can then further analyze** as we do in Section 5. We agree however that the measure has in turn the issues that you mention, and we will explore the direct correlation approach you suggest.
>
>
> ## Calibration
> > The authors make an assumption that better-calibrated models will have lower output entropy. This assumption is false in general [...].
>
> Thanks for pointing this out. We will remove the direct reference to calibration. Our reasoning is as follows: in general, machine prompts are more likely to trigger the correct output and, at the same time, they have lower output entropy. So, while this might not be true for all specific cases, the global trends suggest that machine prompts tend to produce correct answers with more confidence.
>
> ## Table 3 is unclear
>
> > It's hard to figure out what is being correlated with what in Table 3.
>
> *Unless we are mistaken, it seems that there is a typos, and reviewer oqtz is referring to Table 2, and not Table 3.*
>
> We agree on Table 2’s lack of clarity. The first column describes the data used to compute the correlations: human and M-disc prompts, human and M-cont prompts, [...], only human prompts, only M-disc prompts, etc. Then, the next columns provide the correlation between two measures (among input similarity, activation overlap and output agreement). For instance, when considering the set of human and M-disc prompts, we measure a correlation of 0.21 between the input similarity and the activation overlap. The same correlation is 0.72 when considering the set of human prompts only.
>
> Here is an improved version of the Table:
> | Type                 		 | Input predicts Activation?     | Activation predicts Output?     | Input predicts Output?     |
> |---------------------------    |----------------------------    |-----------------------------    |------------------------    |
> | Between Human and M-disc 	 | 0.21 [0.17, 0.25]     		 | 0.01 [-0.04, 0.06]     		 | 0.11 [-0.03, 0,24]		 |
> | Between Human and M-cont 	 | 0.06 [0.03, 0,08]     		 | -0.04 [-0.07, -0.00]   		 | 0.14 [0.04, 0.24] 		 |
> | Between M-disc and M-cont     | 0.06 [0.04, 0.08]     		 | 0.02 [0.00, 0.05]      		 | 0.30 [0.19, 0.40] 		 |
> | Within Human         		 | 0.72 [0.68, 0.75]     		 | 0.65 [0.59, 0.70]      		 | 0.51 [0.43, 0.59] 		 |
> | Within M-disc        		 | 0.84 [0.82, 0.86]     		 | 0.51 [0.48, 0.59]      		 | 0.52 [0.45, 0.59] 		 |
> | Within M-cont        		 | 0.72 [0.68, 0.75]     		 | 0.51 [0.46, 0.58]      		 | 0.61 [0.54, 0.68] 		 |
>
>  We would be grateful to get the reviewer’s opinion on this new version of the table, and if needed, some suggestions for improvement.
>
>
> ## Terminology
> > I think the so-called "human" prompts are mis-labeled [...]. I think "rule-based" or "human-readable" would be a clearer label.
>
> The prompt templates were taken from a number of sources in the literature, as originally collated in the ParaRel resource. The original templates were partially hand-written, partially harvested from the Web, partially manually or automatically paraphrased. One of the human authors of the current paper went through the whole list, filtering out templates that were not well-formed or semantically appropriate, optimizing them for causal model prediction, correcting or paraphrasing other templates, and coming up with further templates by hand. So, **while some of the templates were originally generated by semi-automated procedures, there are also plenty of templates that were hand-generated, and all prompts have been validated by multiple human judges**. In this respect, we think it’s fair to label them as “human” prompts. Note that in principle the templates generated by AutoPrompt or other algorithmic procedures could also be “human-readable” (although that’s not, for the time being, the case).

---

### Official Review · Reviewer_ZpBf · 2023-08-02

**Soundness:** 4

**Excitement:**

4: Strong: This paper deepens the understanding of some phenomenon or lowers the barriers to an existing research direction.

**Paper Topic And Main Contributions:**

Summary: This paper compares how LMs process human generated vs. automatically generated prompts. They build on prior work that has demonstrated the efficacy of automatically generated prompts to investigate how the two types are processed differentially in two LMs. They show that automatically generated prompts tend to be better calibrated than human prompts and activate a different set of (more language-oriented) neurons than human prompts. Overall, I thought this paper was extremely clearly written. While its results feel a bit like “first steps” on a research topic, I believe that they will be of interest to the community and am therefore recommending this paper for acceptance.

**Questions For The Authors:**

(a) (209) Is “micro-accuracy” a term used in the literature? Is there a reference for this? Otherwise I would just use accuracy and qualify what you mean.

(b) (216) Can you provide an example of “[subject][template]”, as I’m not exactly sure what this looks like. Would it be something like “Lavoisier Island continent of Antarctica” ?

(c) (222) How is the entropy measured?

(d) (230) What does this value range over? What counts as “low” and “high” values for this measurement? (More broadly, this section felt a little telegraphic. I would appreciate a little more detail in the final version here.)

(e) (335) It is an interesting result that the prompts converge to similar input representations for larger LMs. Any speculations about why this is going on?

(f) (354) Again, this section felt a little telegraphic. I would appreciate more detail, examples, and motivation for this analysis.

(g) As noted above, what does the error of the measurement look like here? Are the differences between the prompt types big with respect to the error?

(h) (421) What does “resp.” mean?

(j) (520) As noted above, the idea that these connections arose by chance during training seems at odds with the finding from Rakotonirina et al., (2023) that automatic prompts can be made to generalize across LMs. How would you square this hypothesis with their results?

(k) (limitations) I would also add that this work was carried out only in English and that future work could explore this in other languages as well.


**Reasons To Accept:**

(a) Very well written and extremely clear to follow

(b) The authors provide a number of different analyses that point towards the same results, e.g. the two analyses presented in 5.1 and 5.2.

(c) The paper is on a timely topic and does a good job of situating itself within previous work on the subject (at least to my knowledge)

**Reasons To Reject:**

(a) I think some of the analyses could benefit from more careful statistical testing, in particular the analysis in 5.1. It would be good to get a sense of the measurement error for the results reported in Figure 3. In addition, while I do see a slight difference between the human, M-cont and M-disc patterns, these seem relatively weak, compared to the overall pattern which is shared between prompt types, namely that all prompt types clearly have more top neurons in the final layers than, for example, in layers 4, 8, and 12.

(b) One big question I have is how to square the authors' conclusion—that the special pathways triggered by these prompts arose by chance–with the findings from Rakotonirina et al., (2023), indicating that at least some automatically generated prompts can generalize across language models? Given this result, I am not sanguine about this conclusion, and I think the paper could benefit from more discussion on this point.

**Reproducibility:**

4: Could mostly reproduce the results, but there may be some variation because of sample variance or minor variations in their interpretation of the protocol or method.

**Reviewer Confidence:**

4: Quite sure. I tried to check the important points carefully. It's unlikely, though conceivable, that I missed something that should affect my ratings.

**Typos Grammar Style And Presentation Improvements:**

(a) (294) “two order of magnitude” → “two orders of magnitude”

(b) (392) “not only we” → “not only do we”

---

> ### Author Rebuttal · Authors · 2023-08-28
>
> Thank you for your detailed review, with valuable comments and positive feedbacks. Please find the answers of your questions as follows:
>
> ## Statistical analyses
> > [...] some of the analyses could benefit from more careful statistical testing
>
> We have conducted further statistical analyses and intend to include these findings in the revised paper. We appreciate the reviewer's input, as it has significantly enhanced the clarity of our study. Here is a summary of our statistical investigation (limited here to opt350m):
>
> ### Table 1
> We added a 95% confidence interval (95% CI) with respect to the relations/tasks scores (we first averaged the sample-level scores across templates to get relation-level scores. Then we computed the final average scores and their confidence interval based on these relation-level scores).
> | Type   | Micro         	| PPL (10^3)    	| Entropy    	| Attention dist.   |
> |--------|-------------------|-------------------|----------------|-------------------|
> | Human  | 29.5 [11.5, 65.0] | 0.6 [0.1, 1.9]	| 5.0 [3.2, 6.0] | 34.4 [29.2, 39.7] |
> | M-disc | 43.3 [17.0, 79.5] | 40.9 [16.0, 95.0] | 4.3 [1.9, 5.7] | 30.0 [27.3, 32.5] |
> | M-cont | 54.9 [20.7, 86.0] | NaN           	| 2.1 [0.5, 4.3] | 23.2 [21.1, 25.5] |
>
> ### Figure 1
> We provide a bootstrap estimation of the 95% CI (we randomly sample 350 templates 1000 times):
> | Type         		 | Activation overlap     |
> |-------------------    |--------------------    |
> | Human        		 | 65 [63, 67]   		 |
> | M-disc       		 | 31 [30, 33]   		 |
> | M-cont       		 | 45 [44, 46]   		 |
> | Human vs. M-disc 	 | 16 [16, 17]   		 |
> | Human vs. M-cont 	 | 13 [13, 14]   		 |
> | M-disc vs. M-cont     | 15 [15, 15]   		 |
>
> ### Figure 2
> Same as in Fig1:
>
> | Type         		 | Input similarity     |
> |-------------------    |------------------    |
> | Human        		 | 96 [96, 96] 		 |
> | M-disc       		 | 90 [90, 90] 		 |
> | M-cont       		 | 85 [84, 86] 		 |
> | Human vs. M-disc 	 | 86 [85, 86] 		 |
> | Human vs. M-cont 	 | 48 [48, 49] 		 |
> | M-disc vs. M-cont     | 51 [50, 51] 		 |
>
> ### Table 2
> Similarly, we provide the bootstrap 95% CI of the correlations (again we randomly sample 350 templates 1000 times). We also rearrange the table to make it more clear:
>
> | Type                 		 | Input predicts Activation?     | Activation predicts Output?     | Input predicts Output?     |
> |---------------------------    |----------------------------    |-----------------------------    |------------------------    |
> | Between Human and M-disc 	 | 0.21 [0.17, 0.25]     		 | 0.01 [-0.04, 0.06]     		 | 0.11 [-0.03, 0,24]		 |
> | Between Human and M-cont 	 | 0.06 [0.03, 0,08]     		 | -0.04 [-0.07, -0.00]   		 | 0.14 [0.04, 0.24] 		 |
> | Between M-disc and M-cont     | 0.06 [0.04, 0.08]     		 | 0.02 [0.00, 0.05]      		 | 0.30 [0.19, 0.40] 		 |
> | Within Human         		 | 0.72 [0.68, 0.75]     		 | 0.65 [0.59, 0.70]      		 | 0.51 [0.43, 0.59] 		 |
> | Within M-disc        		 | 0.84 [0.82, 0.86]     		 | 0.51 [0.48, 0.59]      		 | 0.52 [0.45, 0.59] 		 |
> | Within M-cont        		 | 0.72 [0.68, 0.75]     		 | 0.51 [0.46, 0.58]      		 | 0.61 [0.54, 0.68] 		 |
>
> > (g) As noted above, what does the error of the measurement look like here [Table 2]? Are the differences between the prompt types big with respect to the error?
>
> We observe that the differences between the prompt types are big with respect to the error, in particular when comparing the correlations measured “within” vs. “between” types.
>
> ### Figure 3
>
> We have incorporated the 95% confidence intervals using the bootstrapping technique.
>
> > It would be good to get a sense of the measurement error for the results reported in Figure 3. In addition, while I do see a slight difference between the human, M-cont and M-disc patterns, these seem relatively weak, compared to the overall pattern which is shared between prompt types, namely that all prompt types clearly have more top neurons in the final layers than, for example, in layers 4, 8, and 12.
>
> As correctly pointed out by Reviewer ZpBf, while the overall trend of having more highly activated neurons in the final layers for all prompt types is evident, the distinctions between prompt types are somewhat less pronounced. Nonetheless, we wish to emphasize that the **differences remain statistically significant, particularly in the first layers (l00 and l04), and to a lesser extent in the final layer.** In these specified layers, we have statistically demonstrated that machine-generated prompts lead to the activation of more neurons compared to human prompts. We will make appropriate revisions to the paper in light of this insight.
>
>
> ## Chance vs. universality?
> > [...] how to square the authors' conclusion—that the special pathways triggered by these prompts arose by chance–with the findings from Rakotonirina et al., (2023), indicating that at least some automatically generated prompts can generalize across language models?
>
> Thanks, this is a great point. When we say that the pathways arose “by chance”, **we do not mean that they arose by random happenstance** (e.g., through lucky initialization seeds), but rather that **they came about as an unintentional side effect of pre-training**. We find it fascinating that, as shown by Rakotonirina and others, very opaque token sequences trigger the same response across different language models. We conjecture that this is due to similarities in architecture (all popular language models have rather similar Transformer backbones) and training data (e.g., parts of the Pile corpus and other Web-derived data are used to train many popular models). Suppose for example that a special pathway arises in response to some quirks in the distribution of the input tokens. Then, two Transformer-based language models exposed to similar data might develop similar pathways. We realize that this is just a wild conjecture. We will discuss it, as such, in the paper.
>
>
> ## Micro accuracy
> > (a) (209) Is “micro-accuracy” a term used in the literature? Is there a reference for this?
>
> The term “micro-accuracy” is from the literature (e.g. in  Zhong et al., 2021). We will make it clearer when revising the paper.
>
>
> ## Template
> > (b) (216) Can you provide an example of “[subject][template]”, as I’m not exactly sure what this looks like. Would it be something like “Lavoisier Island continent of Antarctica”?
>
> An example of “[subject][template]” is:
> Human prompt: *“Lavoisier Island belongs to the continent of”*
> M-disc prompt: *“Lavoisier Island Antarctica submissions Telegraph Travels in”*
>
>
> ## Entropy
> > (c) (222) How is the entropy measured?
>
> The entropy score consists in the average Shannon entropy of the output probability vector computed across all samples of the evaluation set. We will clarify.
>
>
> ## Attention distribution
> (d) (230) What does this value range over? What counts as “low” and “high” values for this measurement? (More broadly, this section felt a little telegraphic. I would appreciate a little more detail in the final version here.)
>
> We quantified the distribution of attention using a method introduced by Ramsauer et al. (2021) (see Figure A.3). This value ranges from 0 to 100. Intuitively, given a row of an attention map of a transformer layer, **it corresponds to the number (in %) of attention values you have to sum to reach 90% of the total attention**. Because attention values are normalized, if the attention is flat then the score will be 90. In contrast, if all the attention is focused on one token, then the score will be close to 1. When we spoke of low and high values, we simply wanted to state that our score decreases as the attention distribution becomes more peaky. We will clarify this.
>
> ## Large LMs' machine prompts convergence
> > (e) (335) It is an interesting result that the prompts converge to similar input representations for larger LMs. Any speculations about why this is going on?
>
> The paper by Rakotonirina you mentioned above found that better-performing machine prompts are, to some degree, more “semantically transparent”, and thus presumably more similar to human prompts. As machine prompt accuracy increases for the larger model (first row of Table 1), it’s possible we are recording a similar phenomenon: **“better” machine prompts that are more transparent, and thus more similar to human prompts**. If this is the case, **still the approximation to human prompts must hit a ceiling at a certain point, as shown by the persistence of effective unnatural language even for ChatGPT-sized models** (e.g.: https://arxiv.org/abs/2307.12507, https://llm-attacks.org/). We will explicitly discuss this in the paper.
>
>
> ## Writting
>
> > (f) (354) Again, this section felt a little telegraphic. I would appreciate more detail, examples, and motivation for this analysis.
>
> We will clarify and reformulate when revising.
>
>
> ## Resp.
> > (h) (421) What does “resp.” mean?
>
> We thought “resp.” is an abbreviation for respectively, but we will avoid it.
>
>
> ## Explore other languages
> > (k) (limitations) I would also add that this work was carried out only in English and that future work could explore this in other languages as well.
>
> That’s a very interesting research direction, and we will add it to current limitations.

---

### Official Review · Reviewer_jWaw · 2023-08-04

**Soundness:** 2

**Excitement:**

2: Mediocre: This paper makes marginal contributions (vs non-contemporaneous work), so I would rather not see it in the conference.

**Missing References:**

The paper misses a significant line of work in uncovering unnatural language processing of Transformer based Language models, including (but not limited to) the following papers:

- Sinha, Koustuv, et al. "Unnatural language inference." arXiv preprint arXiv:2101.00010 (2020).
- Marzoev, Alana, et al. "Unnatural language processing: Bridging the gap between synthetic and natural language data." arXiv preprint arXiv:2004.13645 (2020).
- Pham, Thang M., et al. "Out of order: How important is the sequential order of words in a sentence in natural language understanding tasks?." arXiv preprint arXiv:2012.15180 (2020).
- Gupta, Ashim, Giorgi Kvernadze, and Vivek Srikumar. "Bert & family eat word salad: Experiments with text understanding." Proceedings of the AAAI conference on artificial intelligence. Vol. 35. No. 14. 2021.



**Paper Topic And Main Contributions:**

The paper studies the internal behavior of OPT 350M and OPT 1.3B models with respect to human crafted and machine generated prompts. The paper highlights that while the machine generated prompts (using popular methods such as AutoPrompt and Optiprompt) achieve better scores, they elicit processing behavior which is very different from those of the human prompts. The paper doesn't attempt to answer why such a thing exists, instead proposes analysis methods to highlight the differences in processing with "unnatural inputs", which is the machine generated prompts.



**Questions For The Authors:**

- Question A: It is expected that input similarity would be higher with larger models, as cosine similarity is a poor distance function at high dimensional space. Did the authors try any other distance functions?
- Question B: More simply, can the authors first show the token overlap % between human and m-disc? Would this be highly correlated to the unit activation overlap?
- Question C: Did the authors specifically study the examples where machine generated prompts are outperforming human prompts? Is there anything special about the examples themselves in such cases?
- Question D: When the authors study attention distribution ("attention is focused on a smaller amount of tokens"), is this with respect to the prompt or with respect to the input example? How does the attention distribution over the examples change?
- Question E: The authors note that their evidence suggests the results are more than a happy accident. Did the authors study the logits of the generation, in order to quantify the "accident" in terms of model confidence? As in, does the model becomes more/less confident with machine generated prompts?



**Reasons To Accept:**

- The paper is clearly written with good analysis of the differences in processing human crafted prompts and machine generated prompts. The paper defines the studies well, and come to several insights of how the processing methods employed by the LLMs internally are different, while the model clearly favors the machine generated prompts.



**Reasons To Reject:**

- My main criticism is that the paper essentially does not present anything new or surprising with respect to the model behavior. In Section 5.2, the paper uncovers the top "typical" tokens for each prompt type, and it is noticeably clear how linguistically different the human generated prompt and the machine generated prompts are. Since the surface forms of these prompts are wildly different, it is expected that the internal processing (attention distribution, activation overlap) would also be noticeably different. A better study would have been to isolate the prompts from each category which are mostly similar in their surface forms, and then investigate whether the model prefers one or the other.
- Investigating unnatural language is indeed a very interesting and useful line of work. However, the main question should be _why_ the model is differentiating its processing on these unnatural input rather than human input, which is clearly more abundant in the training data. In this paper, we only get a glimpse of _how_ the model is behaving differently, without any explanation to the performance improvement for unnatural input. The authors do highlight this in the Limitations and I thank them for it. However, since the finding does not improve our understanding towards the "why" problem, the report is not so interesting for the community.



**Reproducibility:**

5: Could easily reproduce the results.

**Reviewer Confidence:**

4: Quite sure. I tried to check the important points carefully. It's unlikely, though conceivable, that I missed something that should affect my ratings.

**Typos Grammar Style And Presentation Improvements:**

- L201: "We end up with 5.9 human" -> "5.9 M human"

---

> ### Author Rebuttal · Authors · 2023-08-28
>
> Thank you for your detailed review, with valuable comments and appreciations. Please find the answers of your questions below:
>
> ## Novelty and interest
>
> > [...] the paper essentially does not present anything new or surprising with respect to the model behavior.
> > Since the surface forms of these prompts are wildly different, it is expected that the internal processing (attention distribution, activation overlap) would also be noticeably different.
>
> Concerning the interestingness of our results, please note that, for each type, we are selecting prompts that are performing the same task and have accuracy above a certain threshold (>10%). Thus, **while the inputs are very different, they produce the same or similar outputs** (note the low correlations between input similarity and output agreement in the top right cell of Table 2). We could thus reasonably expect that there are processing differences in the lower layers, but these differences vanish higher up, where the network starts converging on the output to be generated. **The fact that this is not the case is, in our opinion, highly surprising**. **It suggests that different prompt types produce similar outputs through thoroughly different generation strategies.** The diagnostic classifier experiment in appendix B goes even further, showing that, on each layer, **it is possible to predict prompt type from activation profile even when generalizing across knowledge-retrieval tasks**: that is, the activation profiles of, e.g, the place-of-birth and works-for machine prompts have something in common that allows a shallow classifier that has only seen place-of-birth examples at training time to easily distinguish machine-generated works-for examples from the corresponding human prompts on each layer. Again, we think this is very remarkable.
>
> Concerning the fact that we do not present anything “new”, we respectfully disagree: to the best of our knowledge, despite widespread interest in machine-generated prompts, **ours is the very first mechanistic analysis of how such prompts are processed by a language model**.
>
> We will further clarify all these points when revising the paper.
>
> ## Investigating the *why* instead of the *how*
>
> > Investigating unnatural language is indeed a very interesting and useful line of work. However, the main question should be why the model is differentiating its processing on these unnatural input rather than human input [...].
>
> We fully agree that ultimately the “why” problem is the fundamental one. We think, however, that **we can’t get at the “why” without first characterizing the “how”**. We hope that the set of  “how” results we present can help our community to formulate causal explanations that can already exclude hypotheses that are rejected by our findings (e.g., that differences between prompts only pertain to the earliest processing steps). We are grateful to reviewer oqtz for perfectly capturing our spirit in their review: *“Some readers may find the lack of a causal hypothesis disappointing, but I think for many this will mostly be a hindsight effect thanks to the very clear problem statement in the paper itself: the authors make a strong case that it's worthwhile to study the mechanisms by which prompt engineering works and that too few researchers are currently tackling this question. The detailed exploratory characterization of the empirical phenomenon that this paper provides will likely be an influential early step on the path toward deeper understanding of this question.”*
>
>
> ## Cosine similarity
> > Question A: it is expected that input similarity would be higher with larger models, as cosine similarity is a poor distance function at high dimensional space. Did the authors try any other distance functions?
>
> We are using cosine similarity as we think it is **the most standard measure to compare neural-network-extracted vector representations**. We would be grateful for references about the issues you are mentioning, and we’d be happy to implement any more robust method you recommend. Still, we would be surprised if the large asymmetry in similarity increase between M-cont and human vs. the other comparisons in Fig. 2 was only due to the possible confound you mention (note also that the OPT-350m layer activation space, with its 1024 elements, is already very high dimensional). We have in any case also replicated the analysis using token overlap, as discussed below.
>
>
> ## Token overlap
> > Question B: More simply, can the authors first show the token overlap % between human and m-disc? Would this be highly correlated to the unit activation overlap?
>
> The token overlap between human and m-disc (measured as the intersection over union between the sets of tokens employed by each prompt type) is **2.7%**. As expected, this overlap is low, but this datum might be misleading for at least three reasons:
> 1. Tokens found in M-disc prompts are more **diverse** than those in human prompts: we counted 742 different tokens in M-disc prompts, compared to only 96 in human prompts.
> 2. The very low token overlap **does not explain the significantly positive output agreement** (counting how often two prompt types lead to the same predictions) between Human and M-disc which is 35%.
> 3. The token overlap **does not fully explain activation overlap**.
>
> We assessed (3) by conducting a small experiment with opt350m, which involved measuring the intra-prompt-type token overlap. For each type, we randomly sampled two sets of 50 templates and measured the token overlap between the sets. We repeated this operation 1000 times to obtain an average token overlap within a prompt type. While Human templates exhibited a relatively high overlap (54% token overlap), we observed a low overlap within M-disc templates (7.6%). However, as stated in the article, we demonstrated that M-disc templates tend to activate the same neurons in the LM (as evidenced by an activation overlap of 31%). Consequently, **a low overlap in input tokens does not necessarily imply a low overlap in terms of neuron activation**. While we acknowledge your point that token overlap contributes to the explanation, we firmly believe and provide evidence for the fact that it is imperative to move beyond direct correlations between token and neuron activation overlaps.
>
>
> ## Isolate similar prompts
>
> > A better study would have been to isolate the prompts from each category which are mostly similar in their surface forms, and then investigate whether the model prefers one or the other.
>
> Thanks for the suggestion. It is closely related to what we tried to achieve with the analysis in Table 2, which shows that input similarity (Input in the table) is only weakly correlated to activation overlap and output agreement, suggesting that superficial differences between inputs are not the main driver of processing and output mismatches. Reviewer oqtz also found Table 2 confusing, and we will try to get our message across more clearly when revising the paper. Concerning the analysis you suggest, we are specifically interested in cases where different prompt types lead to similar outputs, as these are the cases where differences in processing might cue genuinely different processing methods. When one prompt type produces the right output and another doesn’t, the processing paths might differ simply because different answers are being produced.
>
> ## Focusing on machine generated prompts outperforming human prompts
> > Question C: Did the authors specifically study the examples where machine generated prompts are outperforming human prompts? Is there anything special about the examples themselves in such cases?
>
> For the reasons we discussed above, **we are focusing on cases where all prompt types lead to relatively good performance**, precisely because we are interested in studying processing when prompts tend to produce the same or similar outputs. Our main question is: *why can these alternative, linguistically “crazy” machine-generated prompts produce coherent outputs comparable to those produced by human prompts for the same semantic tasks?* This is a different question from the one of why machine-generated prompts can even outperform human ones. We however agree that the analysis you recommend is interesting, and we will consider it in future research.
>
>
> ## Attention distribution
> > Question D: When the authors study attention distribution ("attention is focused on a smaller amount of tokens"), is this with respect to the prompt or with respect to the input example? How does the attention distribution over the examples change?
>
> We quantified the distribution of attention using a method introduced by Ramsauer et al. (2021) (see Figure A.3). This method operates at a global level, which means it is computed at the corpus level rather than the local sample level. Consequently, **our measurement is derived with respect to the prompt type and cannot be replicated at the input example level**. While it's possible to modify the method to achieve a local measure of attention distribution, we hold the belief that **the outcome would likely be noisy and challenging to interpret**, particularly given the diversity of templates. As a result, we opted for the more robust approach of using a global measure.
>
>
> ## Quantifying the "accident" in terms of model confidence
> > Question E: The authors note that their evidence suggests the results are more than a happy accident. Did the authors study the logits of the generation, in order to quantify the "accident" in terms of model confidence? As in, does the model becomes more/less confident with machine generated prompts?
>
> We observe that the **output entropy decreases for machine prompts** (Output entropy row of Table 1), suggesting higher model confidence. We will clarify the meaning of this statistic in the revision.
>
>
> ## Typical tokens
> >  In Section 5.2, the paper uncovers the top "typical" tokens for each prompt type, and it is noticeably clear how linguistically different the human generated prompt and the machine generated prompts are
>
> Please kindly note that the tokens of Section 5.2 are not input tokens, but tokens associated to the units activated by the different kinds of prompts.
>
>
>
> ## Additional references
> > The paper misses a significant line of work in uncovering unnatural language processing of Transformer based Language models, including (but not limited to) the following papers:
>
> Thanks for the references, we will incorporate them when revising the paper.

---

### Meta-Review · Area_Chair_FvXy · 2023-09-19

**Recommendation:** 4

**Metareview:**

This paper studies the behavior of models on human-constructed vs machine generated prompts, finding that machine generated prompts are processed in a qualitatively different way compared to human-constructed prompts. The findings in this study are very intuitive, and all reviewers appreciated the analysis in the paper.

---

### Decision · Program_Chairs · 2023-10-07

**Decision:**

Accept-Findings

**Comment:**

This paper studies the behavior of models on human-constructed vs machine generated prompts, finding that machine generated prompts are processed in a qualitatively different way compared to human-constructed prompts. The findings in this study are very intuitive, and all reviewers appreciated the analysis in the paper.